# Cistracurium Besylate 10 mg/mL Solution Compounded in a Hospital Pharmacy to Prevent Drug Shortages: A Stability Study Involving Four Degradation Products

**DOI:** 10.3390/pharmaceutics15051404

**Published:** 2023-05-04

**Authors:** Marine Roche, Cécile Danel, Nicolas Simon, Mostafa Kouach, Myriam Bouchfaa, Christophe Berneron, Pascal Odou, Damien Lannoy

**Affiliations:** 1ULR 7365—GRITA—Groupe de Recherche sur les Formes Injectables et les Technologies Associées, University Lille, F-59000 Lille, France; marine.roche@chu-lille.fr (M.R.);; 2CHU Lille, Institut de Pharmacie, F-59000 Lille, France

**Keywords:** cisatracurium, laudanosine, injectable, drug compounding, drug stability

## Abstract

Background: Stability study of a 10 mg/mL injectable cisatracurium solution stored refrigerated in amber glass ampoules for 18 months (M18). Methods: 4000 ampoules were aseptically compounded using European Pharmacopoeia (EP)-grade cisatracurium besylate, sterile water for injection, and benzenesulfonic acid. We developed and validated a stability-indicating HPLC-UV method for cisatracurium and laudanosine. At each stability study time point, we recorded the visual aspect, cisatracurium and laudanosine levels, pH, and osmolality. Sterility, bacterial endotoxin content, and non-visible particles in solution were checked after compounding (T0) and after M12 and M18 of storage. We used HPLC-MS/MS to identify the degradation products (DPs). Results: During the study, osmolality remained stable, pH decreased slightly, and the organoleptic properties did not change. The number of non-visible particles remained below the EP’s threshold. Sterility was preserved, and bacterial endotoxin level remained below the calculated threshold. Cisatracurium concentration remained within the ±10% acceptance interval for 15 months and then decreased to 88.7% of C0 after M18. The laudanosine generated accounted for less than a fifth of the cisatracurium degradation, and three DPs were generated—identified as EP impurity A, impurities E/F, and impurities N/O. Conclusion: Compounded 10 mg/mL cisatracurium injectable solution is stable for at least 15 months.

## 1. Introduction

Cisatracurium is a non-depolarizing, intermediate-acting, neuromuscular blocker from the benzylisoquinoline family of curares [1,2,3,4,5]. In Europe, cisatracurium is marketed as a besylate salt solution with a concentration ranging from 2 to 5 mg/mL of cisatracurium base [1,5,6]. The drug is used to induce the relaxation of striated muscles during general anesthesia procedures or sedation in the intensive care unit (ICU) [1,3,4,5,7,8]. Cisatracurium is the only curare recommended for prolonged infusion in patients with acute respiratory distress syndromes and in the ICU because it is associated with a lower risk of histamine release [2,8,9]. Due to the massive number of patients requiring invasive ventilation during the coronavirus disease 2019 (COVID-19) pandemic, the demand for neuromuscular blockers increased dramatically worldwide and led to drug shortages [10,11,12,13,14].

In order to be able to manage severe cases of COVID-19 throughout the crisis period, our medical center’s central pharmacy compounded 10 mg/mL cisatracurium solution locally. The formulation was based on those described in the United State Pharmacopoeia (USP) [15] and in the Handbook of Pharmaceutical Manufacturing Formulations [16]. Manufacturers of commercial cisatracurium solutions recommend storage between 2 and 8 °C [3,4,5,7,15] because the drug is reportedly unstable at room temperature [4]. Furthermore, cisatracurium solutions should be stored in the dark and not frozen [3,4,5,7,15].

Two groups of researchers [17,18] have used high-performance liquid chromatographic methods with UV detection (HPLC-UV) methods to study the stability of commercial cisatracurium solutions. Xu et al. studied 2 and 10 mg/mL solutions and found that when stored in glass vials, the drug was stable for at least 45 days at 23 °C and for at least 90 days at 4 °C [17]. When stored in polypropylene syringes, 2 mg/mL cisatracurium solutions were reportedly being stable for at least 30 days at both 4 °C and 23 °C (according to Xu et al. [17]) or at least 7 days at both 5 °C and 25 °C (according to Pignard et al. [18]), and 5 mg/mL solutions were reportedly stable for at least 7 days at both 5 °C and 25 °C [18]. However, there are no literature data on the stability of 10 mg/mL cisatracurium solutions for injection stored in amber glass ampoules at 2 to 8 °C for more than 90 days. When refrigerated and stored in the dark, a commercial 10 mg/mL cisatracurium solution (Nimbex^®^) remained stable for 24 months [5] and lost its potency at a rate of approximately 5% per year [3,4].

Laudanosine is one of the identified end-degradation products (DPs) for cisatracurium; it is produced by both ester hydrolysis and the Hofmann elimination pathway [8,19]. Although laudanosine is thought to have neurotoxic effects [19,20,21,22,23,24,25], it is not commonly quantified as a marker of cisatracurium degradation.

In the context of drug shortages caused by the COVID-19 pandemic, the objectives of the present study were to develop a simple formulation of cisatracurium in solution and to study its stability when refrigerated after local compounding by hospital pharmacies. To this end, we developed and validated an HPLC-UV assay method for cisatracurium and laudanosine and assessed its stability-indicating performance. Lastly, we studied the stability of cisatracurium solutions for injection for up to 18 months and identified the DPs using HPLC-MS/MS. These data prompted us to suggest a degradation pathway for compounded cisatracurium solutions under our storage conditions.

## 2. Materials and Methods

### 2.1. Compounding of Cisatracurium Solutions

Cisatracurium solutions were compounded under ISO 4.8 laminar flow conditions in an ISO 5 aseptic room. European Pharmacopoeia (EP)-grade cisatracurium besylate powder (Lianyungang Guike Pharmaceutical Co., Jiangsu, China) was weighed on a precision scale (Quintix224-1CFR; serial number (SN): 0038103652, accuracy: 0.1 mg in the range 10 mg–220 g; Sartorius, Goettingen, Germany).

The powder was placed in a 47 L stainless steel container (Goavec Engineering, Alençon, France) and dissolved in water for injection (Viaflo 1000 mL Baxter, Guyancourt, France) to obtain a final concentration of 10 mg/mL by stirring at 300 rpm for 20 min with an IKA Eurostar 20 stirrer (EURO ST 20DS500, IKA, Staufen, Germany) and IKA 3 propeller shaft (R1382, IKA). Once a homogeneous solution had been obtained, the pH was adjusted to 3.5 by adding a sufficient quantity (approximately 1.6 g per 20 L) of pharmaceutical-grade benzenesulfonic acid (Sigma Aldrich, Buchs, Switzerland). Thereafter, the pH of a sample was checked by using a calibrated pH meter (HI5222; SN: 02190006991, Hanna Instruments, Woonsocket, RI, USA) and a glass electrode HI1053B (SN: 05292DBN, Hanna Instruments). The solution was then passed through a 0.22 µm Fluorodyne EX EDF KLEENPAK Capsule filter (Ref: NP5LUEDFP1G, PALL, Port Washington, NY, USA) directly into a second 47 L stainless steel container (Goavec Engineering) with Pumpsil^®^ platinum-cured silicone tubing (8.0 mm bore × 1.6 mm wall thickness, part 913.A080.016, Watson Marlow, Falmouth, UK) and a 530S Watson Marlow peristaltic pump (SN: 200608-302166, Ref: 050.9131.10E).

The pharmaceutical raw solution was filtered a second time through a 0.22 µm Fluorodyne EX EDF KLEENPAK Capsule and conditioned in 5 mL type 1 (14.75 EP 0.55) amber glass ampoules (SOTAPHARM, La Ferté Bernard, France), using Pumpsil^®^ platinum-cured silicone tubing and a 530S Watson Marlow filling pump. The ampoules were opened, filled (5.2–5.6 mL per ampoule) and sealed with a ROTA R910 ampoule-filler (SN: 29588/79, ROTA, Wehr, Germany). Each batch contained 4000 5 mL ampoules (extractable volume = 5 mL).

### 2.2. Analytes and Reagents

Cisatracurium chemical reference substance (CRS) and laudanosine CRS were obtained from the European Directorate for the Quality of Medicines and Healthcare (Strasbourg, France).

Chromatographic-grade acetonitrile (ACN; Hipersolv chromanorm, Ref: 83639.320) was obtained from VWR BDH Prolabo (Fontenay-sous-Bois, France), chromatographic-grade methanol (MeOH; LiChrosolv Supelco, Ref: 1.06007.1000) was obtained from Sigma Aldrich (Saint-Louis, MI, USA), and water for injection (WFI; VERSYLENE^®^, sterile water, Ref: B230531) was obtained from Fresenius Kabi France (Sèvres, France). EP-reagent-grade (≥98%) formic acid (Ref: 33015M) and EP-reagent-grade (97%) ammonium formate (Ref: 798568) were obtained from Sigma Aldrich (Saint-Louis, MI, USA).

### 2.3. Apparatus and Optimal Analytical Conditions for the Stability-Indicating Method

#### 2.3.1. Chromatography Conditions

Solutions were analyzed on an Ultimate 3000^®^ chromatography system (Thermo Fisher Scientific, Waltham, MA, USA) composed of a quaternary pump (LPG 3400SD, series #8188086), an automatic sampler maintained at 10 °C (WPS 3000TSL series #8188780), a Peltier effect oven set to 25 °C (TCC 3000 series #06032578), and a diode array detector (DAD3000 series #8188520). Each analytical run lasted 20 min. Data were analyzed using Chromeleon 7 software (version 72.2.6394) and Cobra Wizard and Component Table Wizard algorithms (Thermo Fisher Scientific).

The chromatographic method was based on those described in the literature [15,17,18,26]. The stationary phase consisted of a Hypersil GOLD^®^ C18 precolumn (10 × 4 mm; 5 µm, SN: 20023867, Thermo Fisher Scientific) and a Hypersil GOLD^®^ C18 column (150 × 4 mm; 5 μm, SN: 20058938, Thermo Fisher Scientific). The mobile phase consisted of a mixture of 200 mL of MeOH, 200 mL of ACN, 600 mL of WFI, 10 mL of 98% formic acid, and 10 g of ammonium formate.

The dilution solution (DS) used to dilute samples before injection consisted of a mixture of 200 mL of MeOH, 200 mL of ACN, 600 mL of WFI, and 0.4 mL of 98% formic acid. The DS was distributed into 50 mL vials and stored at −20 °C. For each day of analysis, the required quantities of DS were thawed immediately before use. The detection wavelength was set to 280 nm, corresponding to one of the absorbance peaks of cisatracurium and laudanosine; the other peak at 235 nm is reportedly less specific [15,26]. The flow rate was set to 1.5 mL/min (P ≈ 180 bars). A volume of 20 μL was injected for analysis. The column was maintained at 25 °C, and the autosampler was maintained at 10 °C.

#### 2.3.2. The Forced Degradation Study

A valid stability-indicating method is able to distinguish between the degraded compound and the DPs with a sufficient degree of resolution (resolution factor (Rs) > 1.5) [27,28,29]. To assess our method’s stability-indicating performance, we evaluated the forced degradation of cisatracurium CRS in accordance with the European Society of Hospital Pharmaceutical Technologies-French Society of Clinical Pharmacy (i.e. GERPAC-SFPC) guidelines [27] by mixing equal volumes of a 400 µg/mL cisatracurium solution in DS and 0.005 N NaOH at room temperature, 0.1 N HCl at room temperature, or 30% H_2_O_2_ at 60 °C. The media were neutralized before analysis at 0, 10, 20, and 30 min and 1, 2, and 3 h. Cisatracurium was also degraded by exposing a 400 µg/mL cisatracurium besylate solution to a temperature of 60 °C for three hours or 105 °C for two hours or to UV light (254 nm) for up to 7 days.

Each condition was maintained until the cisatracurium concentration had fallen by 20% [27]. Each sample was diluted to 100 µg/mL in DS and filtered through a 0.22 µm regenerated cellulose acetate membrane filter (Chromoptic, Ref: 17162078, Courtaboeuf, France) prior to analysis.

#### 2.3.3. Assay Validation

Cisatracurium stock solutions were prepared by dissolving precisely 10 mg of cisatracurium besylate CRS (Quintix224-1CFR scale, SN: 0034905650; accuracy: 0.1 mg from 10 mg–220 g, Sartorius) in 10 mL of DS in a volumetric flask. The calibration standards (100, 120, 140, 160, and 180 μg/mL) and validation standards (internal quality controls (IQCs): 110, 130, 150, and 170 μg/mL) for cisatracurium besylate were prepared from these stock solutions. The validation range was chosen so that laudanosine could be assayed from its limit of quantification (LOQ) up to a concentration corresponding to degradation of 10% of the cisatracurium initially present and was based on the hypothesis whereby one molecule of cisatracurium can produce two molecules of laudanosine [19]. Due to the broad concentration range of interest (from 0.1 to 75 µg/mL) and in order to increase the accuracy at low laudanosine concentrations, two calibration ranges were studied: a low range from 0.1 to 2.5 µg/mL and a high range from 2 to 75 µg/mL.

Laudanosine stock solutions were prepared by dissolving precisely 10 mg of laudanosine CRS in 10 mL of DS in a volumetric flask. The stock solutions were diluted in DS to 100 µg/mL (S2) and then 5 µg/mL (S3). The high and low calibration ranges and their corresponding validation standards (IQCs) were prepared from S2 and S3, respectively.

The method was validated in accordance with the French Society of Pharmaceutical Sciences and Techniques (SFSTP) guidelines [30,31,32,33] by three different operators, on three consecutive days, and with different stock solutions for the calibration range (1 calibration per day) and the IQCs (3 batches per day) prepared daily and weighed individually. In accordance with the SFSTP guidelines, the acceptance criteria for the relative standard deviation (for calculations of repeatability and intermediate precision) and relative bias were set to 15% [30,31,32,33].

The SFSTP guidelines are based on “accuracy” (also referred to as “total error”), which is the sum of the systematic error (trueness) and the random error (precision) obtained with validation standards. An accuracy profile is plotted for each analytical method; it shows the total error as a function of the concentration of the validation standard and thus serves as a decision-making tool. The analyst predefines the error tolerance (acceptance limits) and the risk (the β-expectation tolerance interval). Hence, the accuracy profile indicates the probability with which the measurements are included in the acceptance interval.

In accordance with the SFSTP guidelines, we determined the selectivity, response function, linearity, trueness, and precision (repeatability and intermediate precision). The accuracy profile was assessed by taking into account ±10% acceptance limits for quantification (the maximum tolerable error) and a type 1 risk of 5%.

The analytical method’s limit of detection (LOD) and LOQ were calculated according to the International Conference on Harmonization method [34], using the standard deviation of the signal amplitude of a blank solution repeated 10 times and the slope of the response function.

### 2.4. The Stability Study

The stability study was conducted in accordance with the corresponding GERPAC-SFPC guidelines [27]. Forty-five ampoules were immediately analyzed after production (T0): 12 for the non-visible particle count, 20 for the sterility assay, 10 for HPLC-UV quantification, and three for other analyses. The remaining ampoules from the batch were refrigerated (at 2–8 °C) and analyzed (content, pH, and osmolality) after 3 months (M3), M5, M7, M9, M12, M15, M16, M17, and M18. Sterility and bacterial endotoxins were assayed at T0, M12, M15, and M18. Non-visible particles in solution were counted at T0, M5, M9, M12, and M18.

#### 2.4.1. Particle Counting, Osmolality, and pH Studies

At each sampling time point, an operator (always the same) inspected the solutions for a change in color or the presence of a haze, particles (according to the EP 2.9.20 monograph) [35], gas or precipitate, (relative to a WFI control). The pH of the solutions was measured using a pH meter HI5222 (SN: 02190006991, Hanna Instruments) and a glass electrode HI1053B (SN: 05292DBN, Hanna Instruments). Osmolality was measured with an OSMO1 Osmometer (SN 19101407B, Advanced Instruments, Norwood, MA, USA). The measurement was repeated on three ampoules.

Non-visible particles in the preparation were counted (according to the EP method [36]) at the stability studies’ initial time point (T0) and respective final time points (M5, M9, M12, or M18), using an APSS-2000 particle counter (Particle Measuring Systems, Boulder, CO, USA) and SamplerSight Pharma software (version 3.0 SP2 3.0.2.15744, Particle Measuring Systems). To fulfill the analysis’ volume requirements, six ampoules were pooled for each analysis. All analyses were performed twice.

#### 2.4.2. The Sterility Study and the Bacterial Endotoxin Assay

The studies took place in an environmental microbiology laboratory at T0, M12, M15, and M18. The test methodology combined monographs 2.6.1 and 5.1.9. from EP 10.8 [37,38]. The samples were seeded under a vertical laminar flow hood under aseptic conditions. The ampoules’ outside surface was decontaminated with 70° ethanol and covered with a 10 × 10 cm sterile compress. Next, 2.5 mL of infusion solution was injected into the culture broth (heart-brain broth), whose microbial growth promotion had been confirmed previously. The tubes were then incubated at 22 °C and 35 °C for 14 days and observed on Day 4 and Day 14.

In line with the EP [39,40], bacterial endotoxins were assayed with a kinetic–chromogenic method (method D) using EndoSafe^®^—Portable Test System cartridges (Charles River, Charleston, NC, USA) and the corresponding analyzer (Charles River Series 6930). Samples were diluted 800-fold in bacterial-endotoxin-free water (LAL reagent water, Lonza, Walkersville, MD, USA).

#### 2.4.3. HPLC-UV Analysis of Compounded Ampoules

Ampoules were diluted in DS to give a cisatracurium base concentration of 100 μg/mL (equivalent to 134 µg/mL of cisatracurium besylate); this corresponded to the middle of the validated concentration range. Prior to sample analysis, the blanks and IQCs (110, 130, 150, and 170 μg/mL) were prepared under the same conditions. The IQCs were tested in random order. Prior to analysis, each sample was filtered through a 0.22 µm regenerated cellulose acetate membrane filter (Chromoptic, Ref: 17162078).

#### 2.4.4. Apparatus and Analytical Conditions for Identification of the DPs

In order to identify cisatracurium’s DPs at the end of the stability study, samples were analyzed using HPLC-MS/MS on a UFLC-XR device (Shimadzu, Kyoto, Japan) coupled to a QTRAP^®^ 5500 MS/MS hybrid system triple quadrupole/linear ion trap mass spectrometer (AB Sciex, Foster City, CA, USA) equipped with a Turbo V^TM^ electrospray ion source. The instrument was controlled, and data were acquired and processed using Analyst software (version 1.7.2, AB Sciex).

Reverse-phase liquid chromatography (RPLC) separation was carried out on a Hypersil GOLD^®^ column (150 × 4 mm; 5 μm, SN: 20058938) equipped with a guard column (the same stationary phase; 10 × 4 mm; 5 µm, SN: 20023867; Thermo Fisher Scientific) thermostated at 25 °C. The injection volume was either 2 µL or 20 µL. The sample was eluted at a flow rate of 1 mL/min in isocratic mode. The mobile phase consisted of 5 mM ammonium formate and formic acid 0.1%/MeOH/ACN—60/20/20 (%*v*/*v*/*v*). Each analysis took 12 min. The mass spectrometer was operated in positive ionization mode. The ion source parameters were optimized and set as follows: ion spray voltage (5500 V), nebulizer gas flow (air, 50 psi), curtain gas flow (nitrogen, 25 psi), source temperature (600 °C) with the auxiliary gas flow (air) set to 50 psi, and declustering potential (DP, 100 V). The mass spectrometer was operated at a unit resolution. The full scan mode was monitored for *m*/*z* values of between 300 and 1000 Da. In the MS/MS analysis, the product ion scan was used, and the spectra were recorded for *m*/*z* values of between 50 and 600 Da. The collision energy was optimized for each compound.

#### 2.4.5. Preparation of Fractions for the Identification of Cisatracurium DPs Using LC-MS/MS

Each DP or cisatracurium fraction was collected manually (using semi-preparative HPLC-UV but without changing the scale of the purification) prior to the LC-MS analysis. Indeed, increasing the injected volumes or concentrations would have worsened the peak resolution. Next, 10 samples of cisatracurium solution from an ampoule having been stored for 18 months at 2–8 °C were successively injected. The 10 separate fractions were pooled, and the obtained volumes (>1 mL) of each DP solution were sufficient for LC-MS analysis. The DPs’ concentrations were sufficient because reliable MS signals were obtained; evaporation and reconstitution were therefore unnecessary.

#### 2.4.6. Data Analysis

Chemical stability was considered to be acceptable if the remaining concentration of cisatracurium was between 90 and 110% of the initial value (C0) and devoid of toxic DPs or any significant variations in other parameters studied (organoleptic characteristics, pH, osmolality, etc.). There are no specific literature data on the toxicity of cisatracurium’s DPs other than laudanosine. Given that cisatracurium’s DPs do not have neuromuscular blocking activity [18], we considered that semi-quantitative monitoring was sufficient for DPs other than laudanosine. According to the USP monograph on cisatracurium besylate injection, the acceptance criteria for DPs is 4% of the cisatracurium (base) area for laudanosine, 4.3% for cis-quaternary acid (EP impurity A [26]), 5% for cis-quarternary alcohol (EP impurity F [26]), 2.5% for cis-monoacrylate (EP impurity O [26]), and 14.4% for total DPs [15].

With regard to non-visible particle counts in solutions for parenteral use, the 2.9.19 EP monograph’s maximum authorized value is 6000 particles/container for particles >10 μm and 60 particles/container for particles >25 μm [36].

The bacterial endotoxin threshold was calculated for an average bodyweight of 70 kg, according to the EP monograph’s formula ($Endotoxin\;limit=\frac{K}{M}$) [39,40] and with a threshold pyrogenic dose (K) of 5 IU endotoxin per kg body mass and a maximum total dose administered of 20 mg/h (based on our ICU’s standard operating procedures [9,41,42,43]; the resulting value was 17.5 IU/h (8.75 IU/mL).

## 3. Results and Discussion

### 3.1. The Forced Degradation Study

Four DPs were detected during the forced degradation study: one was clearly laudanosine (retention time (RT) = 1.7 min) and the other three were denoted as DP1, DP2, and DP3 (with RTs of 1.4, 2.1, and 8.0 min, respectively). The chromatograms from the forced degradation assays are shown in Appendix A.

At T0, 87% of the cisatracurium had been degraded by the addition of 0.005 N NaOH (Appendix A); this led to strong increases in the levels of the four DPs. After 10 min of exposure, the cisatracurium had been fully degraded, and only DP1 and laudanosine were detected.

Cisatracurium was slightly degraded by the addition of 1N HCl (Appendix A); 13.8% had been degraded after 3 h of exposure, and the levels of DP1 and DP2 increased.

When exposed to a temperature of 105 °C for two hours (Appendix A), 27% of the cisatracurium was degraded—mainly into laudanosine and DP3. No degradation was evidenced after 3 h at 60 °C (0%).

Cisatracurium is fairly stable in an oxidizing environment (Appendix A); only 13% had been degraded after 2 h of exposure to 30% H_2_O_2_ at 60 °C, and the levels of laudanosine and DP3 increased.

When exposed to UV light (254 nm) (Appendix A), about 5% of the cisatracurium was degraded every 24 h. Hence, 19% had been degraded after 96 h. We identified two new major DPs (RTs = 4.5 min and 5.5 min) and two minor DPs (RTs = 9.5 min and 11 min).

These results are consistent (in part) with those reported by Xu et al. [17] and Pignard et al. [18], who found that cisatracurium degraded under acidic conditions (HCl 1 N), alkaline conditions (NaOH 1 N), and oxidizing conditions (H2O2 3%) [17], and after heat exposure [17,18]. Based on the literature data, the researchers evidenced two DPs: laudanosine and monoquaternary acrylate [18]. Nevertheless, in view of the chromatogram presented in the reports, two other (unidentified) DPs were also present. These findings indicate that our stability-indicating method is valid because there was no interference between the cisatracurium peak and the DPs’ peak (Rs >1.5) [27,28,29].

To assess the specificity of our method, we performed LC-MS/MS experiments (see below).

Our cisatracurium compounding and storage techniques were appropriate with regard to the degradation observed under forced conditions. To ensure the cisatracurium’s stability, we adjusted the pH to 3.5, filter-sterilized the solution twice (in preference to heat sterilization), and stored it in amber glass ampoules (for UV protection).

### 3.2. Validation of the Assay Methods

#### 3.2.1. The Cisatracurium Assay

The intra-day precision (random error), inter-day precision (intermediate precision), and relative bias were below 2% (Table 1). The regression line parameters for cisatracurium are summarized in Table 2. The 95% accuracy profile was within the predefined acceptance limits for cisatracurium besylate concentrations ranging from 110 to 170 µg/mL (Figure 1).

#### 3.2.2. The Laudanosine Assay

The intra-day precision (random error), inter-day precision (intermediate precision), and relative bias were below 3% (Table 3). The regression line parameters for low and high laudanosine concentration ranges are summarized in Table 4. The 95% accuracy profile was within the predefined acceptance limits for both low and high concentration ranges (Figure 2A,B).

### 3.3. The Stability Study

Over an 18-month period, we studied the stability of 10 mg/mL cisatracurium besylate solutions compounded in the hospital pharmacy and stored at 2–8 °C in amber glass ampoules.

#### 3.3.1. Particle Counts, Osmolality, and pH

Throughout the stability study, no changes in the cisatracurium solution’s organoleptic properties were noted. The mean ± standard deviation (SD) osmolality remained stable (and within the validated range of 10 to 30 mOsm/kg) throughout the study: 25.0 ± 0.0 mOsm/kg at T0, 24.6 ± 0.8 mOsm/kg at M15, and 24.7 ± 0.7 mOsm/kg at M18 [15,44]. The cisatracurium solution’s pH decreased slightly from 3.43 ± 0.02 (T0) to 3.14 ± 0.01 at M15 and 3.10 ± 0.02 at M18 but remained within the validated range (from 3.0 to 3.7) [15,44]. Although the number of non-visible particles ≥ 10 µm decreased slightly, the number of non-visible particles ≥ 25 µm increased slightly but remained well below the EP threshold for injectable solutions [36] (Figure 3A,B). The measurements are described in detail in the Appendix A.

#### 3.3.2. The Sterility Assay and the Bacterial Endotoxin Assay

No colonies grew in cisatracurium solutions stored at between 2 and 8 °C under any of the test conditions; we conclude that the solutions remained sterile for up to 18 months. The level of bacterial endotoxin was below 8 IU/mL at each time point and thus was below the calculated threshold of 8.75 IU/mL [39,40].

#### 3.3.3. The HPLC-UV Analysis of Compounded Ampoules

The mean ± SD cisatracurium concentration immediately after compounding was 9.73 ± 0.20 mg/mL.

The cisatracurium concentration decreased slightly during the first 90 days of storage and then stabilized within the predefined acceptance interval (Figure 4). From M15 onwards (at a cisatracurium concentration equal to 93.9% of C0), the concentration fell steadily and went below the lower boundary of the ±10% acceptance interval; the concentration was 88.7% of C0 after 18 months of storage at between 2 and 8 °C. Based on these results, we conclude that refrigerated ampoules of cisatracurium solution are stable for 15 months. The fact that our initial cisatracurium concentration, for the unique pilot batch produced for stability study, equals 97% of the rated concentration leads to a reduced shelf life as it will reach the limit of 90% of the nominal concentration earlier than with a 100% initial concentration.

Because the rate of lost potency is constant [3,4], this issue could be overcome by using an overage in our process of compounding ampoules, as it is mentioned in the Nimbex^®^ drug product information [45]. Even if the overage is usually discouraged according to ICH Q8 R2 [46], in our case, it is an interesting add-on initiative that needs to be considered, even if it needs to accurately define a potential amount justification for this drug with a very low therapeutic margin.

Additionally, a stability study should be performed to assess the shelf life of cisatracurium ampoules with a defined overage.

We estimated the shelf life by applying (i) the GERPAC-SFPC methodology and (ii) the ICH (Q1E) model with a linear fit and confidence intervals. The shelf life was defined as the intercept between the lower confidence interval and the 90% assay value [47]. Using the ICH Q1E criteria, the estimated shelf life was 16.3 months (Appendix A), this is consistent with our other results.

To the best of our knowledge, the present study is the first to have demonstrated that a refrigerated cisatracurium solution can be stable for more than 90 days [17,18].

The laudanosine concentration increased from 15.24 ± 1.34 µg/mL at T0 to 61.09 ± 0.85 µg/mL at M15, which corresponds to degradation of about 1.5% of the initial amount of cisatracurium. Although the laudanosine peak area was 5 times greater after M18 than at T0, the peak areas of DP1 (RT = 1.5 min), DP2 (RT = 2.1 min), and DP3 (RT = 8.0 min) increased 50-fold, 30-fold and 6-fold, respectively, over the same period (Figure 5). These results suggest that in contrast to the commonly accepted hypothesis, Hofmann degradation of cisatracurium into two molecules of laudanosine is not the major degradation pathway. Indeed, other DPs (especially DP1 and DP2) appeared to be generated to a greater extent than laudanosine during our stability study (Figure 5).

The total DP peak areas corresponded to 6.68% and 8.25% of the cisatracurium base peak area at M15 and M18, respectively; these values were therefore below the 14.4% acceptance limit set by the USP [15]. Since chemical reference substances for cisatracurium DPs are not available, we cannot quantify them. In view of potential differences in molar absorbance between DPs, we cannot draw conclusions about the mass balance by using the DPs’ %area. Nevertheless, if we compare the %area loss of cisatracurium between T0 and M15 or M18 with the %area increase in DPs over the same period, the difference is approximately ±2%.

Cisatracurium diastereoisomers would have been detected by our HPLC-UV method; indeed we analyzed the atracurium CRS and found three different retention times for the trans-trans, cis-trans, and cis-cis diastereoisomers.

To confirm the cisatracurium solution’s stability after 15 months, we sought to identify DP1, DP2, and DP3 and check that their peak areas (expressed as a percentage of the cisatracurium (base) peak area) were below the USP acceptance limits [15]. Hence, we used LC-MS/MS to identify the DPs. Our identification of the DPs also prompted us to suggest a degradation pathway for compounded cisatracurium solutions stored at between +2 and +8 °C.

#### 3.3.4. Identification of Cisatracurium DPs, Using LC-MS

The mobile phase typically used for HPLC-UV analysis was not compatible with MS analysis and so was replaced with a less salt-loaded phase. The chromatographic profiles of the MS-compatible mobile phase and the standard HPLC-UV mobile phase were broadly similar but differed with regard to some of the RTs. Our strategy was then to isolate the DPs by semi-preparative HPLC-UV, prior to their analysis using LC-MS. Four fractions corresponding to cisatracurium DPs and one fraction corresponding to cisatracurium were collected, according to the HPLC-UV chromatographic profile (Figure 6).

This LC-MS analysis enabled us to identify the DPs by comparison of the *m*/*z* values with the masses of the impurities described in the EP cisatracurium monograph [26]. We identified EP impurity A (*m*/*z* = 430, M^+^) in fraction 1 (DP1, Figure 7A), EP impurity E and/or stereoisomer F (*m*/*z* = 516, M^+^) in Fraction 3 (DP2, Figure 7B), and EP impurity N and/or stereoisomer O (*m*/*z* = 570, M^+^) in Fraction 4 (DP3, Figure 7C). The LC-MS analysis also showed that the compound in fraction 2 was laudanosine (*m*/*z* = 358, M + H^+^); these results confirmed the previous analysis in which the RT was the same as for laudanosine CRS (Figure 7D). For greater specificity, we carried out LC-MS/MS experiments. The main fragments in each fraction are shown in Table 5; the results confirmed the correspondences between the DPs and the EP impurities and/or stereoisomers [48].

The LC-MS/MS analysis of the fraction corresponding to cisatracurium’s RT (5.0 min) confirmed (i) the absence of cisatracurium-DP co-elution and (ii) the specificity of the previously developed HPLC-UV method (Figure 8, Table 5). These results confirmed the stability of cisatracurium solutions at M15. Indeed, the peak areas for EP impurity A (DP1), laudanosine, EP impurity E and stereoisomer F (DP2), and EP impurity N and stereoisomer O (DP3) respectively accounted for 2.7%, 0.8%, 2.7%, and 0.8% of the peak area for cisatracurium (base) at the corresponding degradation time point; the values for the DPs were therefore below the USP threshold [15] (Table 6). Our LC-MS method is not suitable for the detection of acrylic acid, which might potentially be generated by the degradation of DP3.

The fact that the laudanosine generated during the 18-month stability study accounted for less than a fifth of the cisatracurium degradation indicated that another degradation pathway was involved. In view of the nature of the DPs and changes over time in the corresponding peak areas (Figure 5), we believe that cisatracurium was mainly degraded by the ester hydrolysis pathway (generating impurities A and E and/or stereoisomer F, which occur in both acid and alkaline environments) [19]. Hofmann degradation (a pathway favored by heat exposure) of cisatracurium occurred to a lesser extent and led to the formation of laudanosine and impurity N (and/or stereoisomer O) [19]. The putative cisatracurium degradation pathways are shown in Figure 9. Our present results are not consistent with those of Blazewicz et al. [19], who observed Hofmann degradation only. This discrepancy might be due to interstudy differences in the operating procedures: Blazewicz et al. examined pharmaceutical preparations of cisatracurium diluted to 100 µg/mL in 0.1% formic acid in water or in 20% methanol.

## 4. Conclusions

With regard to physicochemical characteristics, pH, osmolality, sterility, and bacterial endotoxin levels, we confirmed that a 10 mg/mL cisatracurium solution is stable for at least 15 months when stored at between 2 and 8 °C in amber glass ampoules. This finding suggests that cisatracurium solutions can be safely compounded in hospital pharmacies in the event of shortages (such as those experienced during the COVID-19 pandemic). Our results confirmed previous reports on cisatracurium degradation. However, in contrast to the outcomes of forced degradation assays in vitro, we found that the ester hydrolysis pathway was more active than the Hofmann elimination pathway at 2 to 8 °C; this raises the question of whether laudanosine should be used as the sole marker of cisatracurium degradation in stability studies [18]. A stability study of cisatracurium solutions stored at room temperature might usefully complement the present work because the degradation pathways might differ from those observed here at between 2 and 8 °C.

## Figures and Tables

**Figure 1 pharmaceutics-15-01404-f001:**
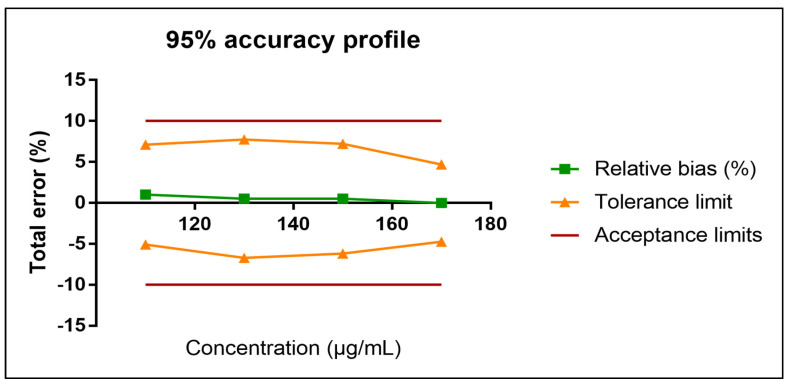
The 95% accuracy profile for cisatracurium.

**Figure 2 pharmaceutics-15-01404-f002:**
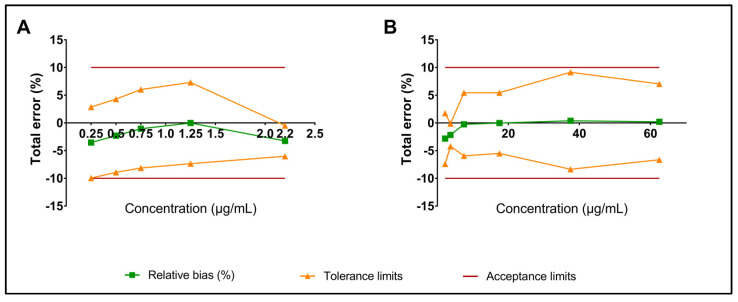
The 95% accuracy profile for laudanosine. (**A**) Low concentrations range. The lower tolerance limit for a 0.25 µg/mL concentration was −9.95%; (**B**) High concentrations range.

**Figure 3 pharmaceutics-15-01404-f003:**
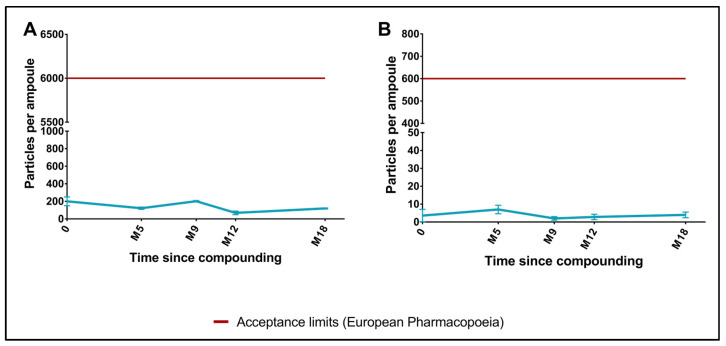
Changes over time in non-visible particle counts (per container) for 10 mg/mL cisatracurium solutions stored between 2 and 8 °C. (**A**) Non-visible particles ≥ 10 µm; (**B**) Non-visible particles ≥ 25 µm.

**Figure 4 pharmaceutics-15-01404-f004:**
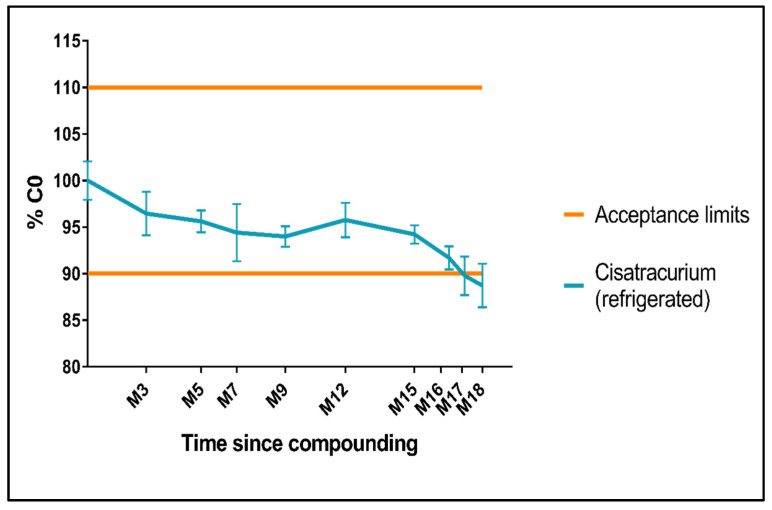
Changes over time in the cisatracurium concentration, as a percentage of C0. C0 was 9.73 ± 0.20 mg/mL.

**Figure 5 pharmaceutics-15-01404-f005:**
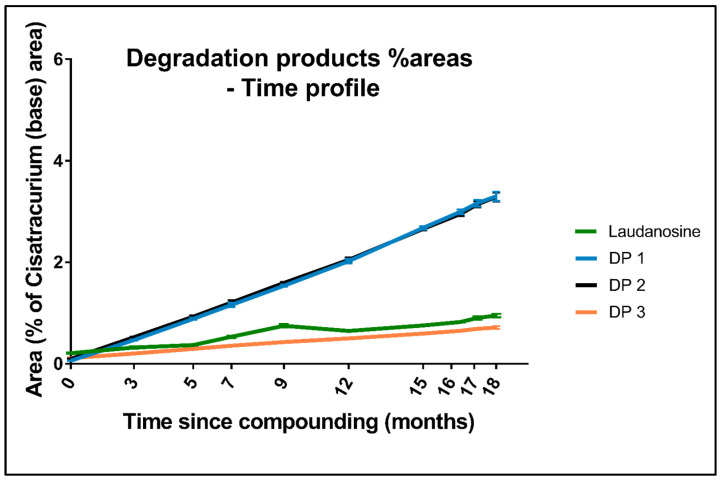
Changes over time in the peak % areas of cisatracurium DPs.

**Figure 6 pharmaceutics-15-01404-f006:**
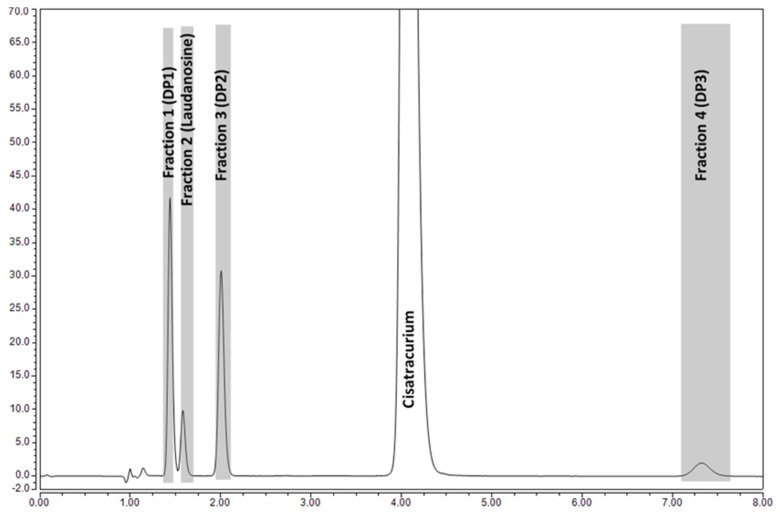
Chromatogram of the cisatracurium solution after storage for 18 months at 2–8 °C. The four collected fractions for DPs are indicated in gray.

**Figure 7 pharmaceutics-15-01404-f007:**
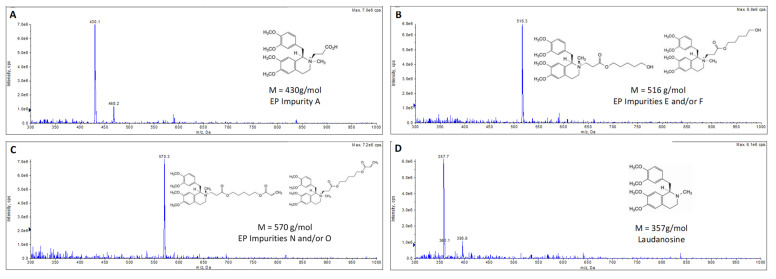
LC-MS analysis of fractions collected using the HPLC-UV-validated method. The spectra were processed by subtracting the mobile phase spectrum: (**A**) Fraction 1: DP1, corresponding to European Pharmacopoeia impurity A (*m*/*z* = 430, M^+^) [26]. (**B**) Fraction 3: DP2, corresponding to European Pharmacopoeia impurity E and/or stereoisomer impurity F (*m*/*z* = 516, M^+^) [26]. (**C**) Fraction 4: DP3, corresponding to European Pharmacopoeia impurity N and/or stereoisomer impurity O (*m*/*z* = 570, M^+^) [26]. (**D**) Fraction 2: laudanosine, corresponding to European Pharmacopoeia impurity C (*m*/*z* = 358, M + H^+^) [26].

**Figure 8 pharmaceutics-15-01404-f008:**
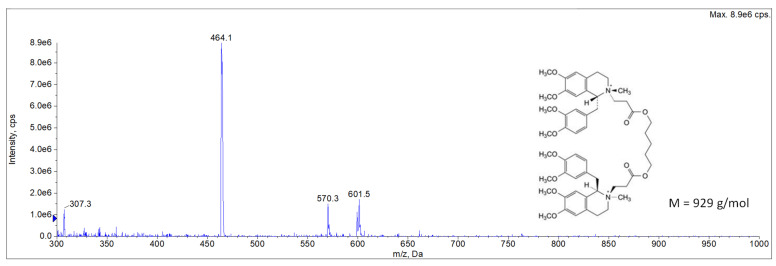
LC-MS analysis of the fraction corresponding to cisatracurium *m*/*z* = 464, M^2+^ (molecular weight = 929 g/mol) [26]. The spectrum was processed by subtracting the mobile phase spectrum. *m*/*z* = 570, M^+^ and *m*/*z* = 601, M^+^ corresponds to cisatracurium fragments generated in the ion source.

**Figure 9 pharmaceutics-15-01404-f009:**
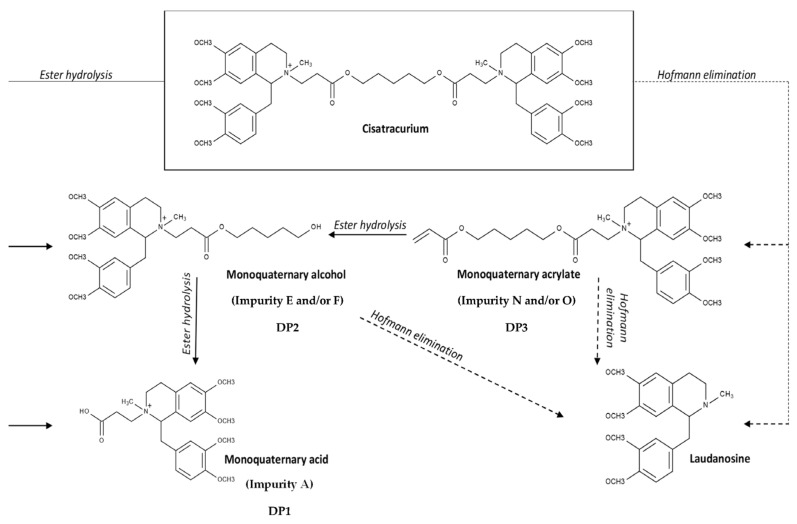
Summary of degradation pathways for cisatracurium and structures of the DPs (adapted from [19]. Impurities A, E, F, N, and O were named to match those described in the European Pharmacopoeia (E and F are stereoisomers, as are N and O).

**Table 1 pharmaceutics-15-01404-t001:** Relative standard deviations (RSDs, %) for repeatability, intermediate precision (IQCs), and relative bias (%) (cisatracurium assay).

Compound	Concentration (µg/mL)	%RSD Repeatability	%RSD Intermediate Precision	Relative Bias (%)
Cisatracurium (besylate salt)	110	0.56%	1.68%	1.01%
	130	0.71%	1.99%	0.51%
	150	0.82%	1.39%	0.51%
	170	0.21%	1.29%	−0.02%

**Table 2 pharmaceutics-15-01404-t002:** Regression line parameters for cisatracurium, including the LOD and the LOQ.

Compound	Concentration Range (µg/mL)	R²	Slope	Intercept (µg/mL)	LOD (µg/mL)	LOQ (µg/mL)
Cisatracurium (besylate salt)	100–180	0.9989	0.117	−0.667	0.78	2.38

**Table 3 pharmaceutics-15-01404-t003:** Relative standard deviation values (%) for repeatability, intermediate precision (IQCs), and relative bias (%) (laudanosine assay).

Laudanosine	Concentration (µg/mL)	%RSD Repeatability	%RSD Intermediate Precision	Relative Bias (%)
Low concentration range	0.25	2.64%	2.64%	−3.54%
	0.5	1.04%	1.85%	−2.31%
	0.75	0.80%	1.96%	−1.05%
	1.25	0.89%	2.03%	−0.02%
	2.2	0.57%	0.89%	1.07%
High concentration range	2.2	0.57%	1.27%	−2.82%
	3.7	0.45%	0.66%	−2.17%
	7.5	0.78%	1.58%	−0.25%
	17.5	0.76%	1.52%	−0.02%
	37.5	0.36%	1.77%	0.39%
	62.5	0.37%	1.38%	0.20%

**Table 4 pharmaceutics-15-01404-t004:** Regression line parameters for low and high laudanosine concentration ranges, including the LOD and the LOQ.

Laudanosine	Concentration Range (µg/mL)	R²	Slope	Intercept (µg/mL)	LOD (µg/mL)	LOQ (µg/mL)
Low concentration range	0.1–2.5	0.9996	0.213	0.008	0.290	0.877
High concentration range	2–75	0.99998	0.213	0.006	0.290	0.878

**Table 5 pharmaceutics-15-01404-t005:** Identification of cisatracurium DPs, using LC-MS/MS. DP: degradation product. Product ions are classified according to their relative abundance. MS/MS spectra are displayed in Appendix A.

Compound	Impurity	Parent Ion (*m*/*z*)	Product Ions (*m*/*z*)	Collision Energy (eV)
Cisatracurium	-	464 (M2+)	189; 307; 151; 327; 165; 601; 370; 516	30
DP1	A	430 (M+)	116; 189; 151; 165; 327; 370	35
Laudanosine	C	358 (M+H+)	260; 206; 189; 327	25
DP2	E and/or F	516 (M+)	202; 116; 189; 370; 151; 165	40
DP3	N and/or O	570 (M+)	256; 189; 370; 151; 516	40

**Table 6 pharmaceutics-15-01404-t006:** Peak areas for cisatracurium DPs (expressed as a percentage of the cisatracurium (base) peak area at the corresponding degradation time point) after 15 and 18 months of storage at between 2 and 8 °C.

Degradation Product [26]	EP Impurity A	Laudanosine	EP Impurity E (or F)	EP Impurity N (or O)	Total DPs
M15	2.7%	0.8%	2.7%	0.6%	6.7%
M18	3.3%	0.9%	3.3%	0.7%	8.2%
USP acceptance limit [15]	4.3%	4%	5%	2.5%	14.4%

## Data Availability

All data are already available in Appendix A.

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
