# Peer review of "Cistracurium Besylate 10 mg/mL Solution Compounded in a Hospital Pharmacy to Prevent Drug Shortages: A Stability Study Involving Four Degradation Products"

_pharmaceutics, 2023, doi:10.3390/pharmaceutics15051404_

Round 1
Reviewer 1 Report
The manuscript deals with the stability study of Cistracurium besylate 10 mg/mL solution compounded in a hospital pharmacy to prevent drug shortages.
The study is interesting and well organized.
I suggest paying attention to some inaccuracies (line 37; line 82-83).
Line 249-251: sentence should be moved to Results section.
Line 308: the reader would be helped indicating minutes, instead of the abbreviation T10.
Table 2 and 4: please, merge into one table.
Reviewer 2 Report
Overall impression: The article is well written, the topic is significant (or at least was during peak covid time). Compared to other articles on compounded or in-use stability, this one is above average. I do have some reservations on stability evaluation and conclusion, see details below.
General comments: evaluation of stability data should be improved by applying ICH methodology (linear curve fitting, application of uncertainty interval), which would shorten the shelf life from the currently proposed 15 months (to account for uncertainty). This is especially so since only one batch was tested. Also the assay given as % label should be taken into account and the possibility of including an overage at least discussed. A commercial product with identical composition to the one in this article exists in other markets (at least US) and should be referenced and discussed (Nimbex 10 mg/mL) with respect to storage condition, stability data and shelf life.
Specific comments:
overage in nimbex is indicated here: Nimbex · Abbvie Corporation · 8401 Trans-Canada Highway, Saint-Laurent, Quebec H4S 1Z1, CANADA (opengovca.com)
line 44: add do not freeze and protect from light. Refrigerated condition range is usually 2-8°C (see ICH), not 4-8°C as repeatedly written in the manuscript.
Line 59: One could argue that since 80% of cisatracurion is metabolised into laudanosine in vivo (NIMBEX [package insert]) that the presence of laudanosine in the drug product is of lesser concern
line 81: add the (approximate) amount of added benzenesulphonic acid
line 85: change to kleenpak (on several occurence)
line 91: state the volume of solution filled in the ampule (was it equal to the nominal volume of the ampule?). You could discuss whether one could use here vials with rubber stoppers to improve generizability.
line 279: stability evaluation should be based in ICH criteria (Q1E) namely linear fit, confidence interval calculation and the shelf life established where the lower CI reaches 90% assay value. Batch variation (assay deviating from 100% of the labelled 10 mg/mL) would additionally shorten the shelf life and this should be discussed. Batch assay values should also be given (at least at time=0) as % label or absolute concentration mg/mL.
line 375: please add a table of stability data for all parameters across all stability timepoints. You could expand table 6. This allows you to remove the stability data from text which would make it much more reader-friendly. Fig3 could be omitted to save space. Include the impurity data in units % (or area%)
line 403: change the stability conclusion based on the outcome of ICH stability data evaluation.
line 404-409: this is repetition of the text in 46-55 and should be removed or shortened. Nimbex commercial product data also includes stability.
Fig 4: add linear trendline with uncertainty interval, discuss how the 97% initial assay (97% of the label) affects the shelf life. Compare the degradation rate based on linear trend to the literature citation for nimbex (5% decrease per year when refrigerated)
Fig5: change units to % or area%. Remove M from x axis labels and add unit (months) into x axis name. Discuss the mass balance (does assay decrease result in an equal impurity increase)
line 480: Discuss the possibility of acrylic acid formation (which would not be detected by the impurity method) and possibility of cisatracurium diastereomers formation (see european pharmacopea monograph).
line 493: should the m/z read 464.5, rounded to 465?
figure 9: some text is not seen.
line 528: change according to a revised stability data evaluation (ICH)
Reviewer 3 Report
Authors developed and validated a stability-indicating HPLC-UV method for cisatracurium and laudanosine.
The study is very accurate and results are clearly presented and adequately discussed. The scientific soundness is of good level.
Round 2
Reviewer 2 Report
Thank you for considering my comments;
the remaining issues are:
- potential for overage still not discussed in the article (the authors have included an overfill, which is ok). For drugs with significant degradation, an overage might be reasonable (intentionally increasing the concentration a few percentage above the label concentration).
- th ICH evaluation of data with a linear fit is ok, it is not uncommon to have rsquared in the range 0.7-0.8 and this is fine. No timepoints should be discarded, as they represent the actual variability (unless there was an assignable cause for disqualifying these values). I would strongly recommend replacing figure 4 with figure 1 from your response, as it is a result of a more formal statistical data treatment according to ICH.
At 5% degradation per year, the 97% initial assay would decrease the shelf life by about 3/5 years and this should be discussed in the article.
Consider adding your response on mass balance and diastereomers into the actual article.
